# Effects of Dietary *Bacillus subtilis* HC6 on Growth Performance, Antioxidant Capacity, Immunity, and Intestinal Health in Broilers

**DOI:** 10.3390/ani13182915

**Published:** 2023-09-14

**Authors:** Shun Liu, Gengsheng Xiao, Qi Wang, Qingyang Zhang, Jinpeng Tian, Weifen Li, Li Gong

**Affiliations:** 1Guangdong Provincial Key Laboratory of Animal Molecular Design and Precise Breeding, School of Life Science and Engineering, Foshan University, Foshan 528225, China; ls70152022@163.com (S.L.); gsxiao2021@163.com (G.X.); qyzhang17814657238@163.com (Q.Z.); tian06062022@163.com (J.T.); 2College of Animal Science, Zhejiang University, Hangzhou 310058, China; 13430230952@163.com (Q.W.); wfli@zju.edu.cn (W.L.)

**Keywords:** *Bacillus subtilis*, growth performance, antioxidation, immune response, gut microbiota

## Abstract

**Simple Summary:**

In the post-antibiotic era, the attainment of higher feed efficiency has become the primary goal of the poultry farming. *Bacillus subtilis* is commonly used as a substitute for antibiotics in animal and poultry feed. The results demonstrated that dietary 5 × 10^8^ cfu/kg *Bacillus subtilis* HC6 increased the feed efficiency, antioxidant capacity, and mRNA expression of pro-inflammatory cytokines in the jejunal mucosa, and decreased the activity of diamine oxidase in serum, which might be attributed to the modulation of community composition and functions of cecal microbiota in white-feathered broilers. Our results provide new insights and evidence for the application of probiotics in broiler breeding.

**Abstract:**

This study aimed to investigate the impact of *Bacillus subtilis* HC6 on the growth performance, immunity, antioxidant capacity, and intestinal health of broilers. A total of 180 one-day-old white feather broilers were randomly divided into two experimental groups, each comprising six replicates of fifteen chicks from 1 to 50 d of age. The groups were either fed a basal diet (CON) or the same diet supplemented with 5 × 10^8^ cfu/kg of *Bacillus subtilis* HC6 (BS). Our results indicated that compared with the CON, dietary supplementation with BS increased feed efficiency during d 21–50 and d 1–50 (*p* < 0.05). Moreover, BS supplementation enhanced antioxidant capacity in the serum and liver, and also decreased the activity of diamine oxidase and the level of endotoxins (*p* < 0.05). Additionally, BS treatment increased the villi height in the jejunum and ileum, increased the ratio of villus height/crypt depth in the ileum, upregulated the expression of tight junction proteins in the jejunal mucosa, and downregulated the levels of IL-22 and IFN-γ on day 50 (*p* < 0.05). Principal coordinates analysis yielded clear clustering of two groups; dietary BS increased the relative abundance of *Bacteroidales_unclassified* (genus) and *Olsenella* (genus), and decreased the abundance of genera *Alistipes* on day 50, which identified a strong correlation with FCR, serum differential metabolites, or differential gene expression in the jejunal mucosa by spearman correlation analysis. The PICRUSt2 analysis revealed that supplementation with BS enriched the pathways related to xenobiotics biodegradation and metabolism, carbohydrate metabolism, energy metabolism, signaling molecules and interaction, the digestive system, and transport and catabolism. These results demonstrated that dietary BS increased feed efficiency, antioxidant capacity, and the mRNA expression of pro-inflammatory cytokines in the jejunal mucosa; and decreased the activity of diamine oxidase in serum, which might be attributed to the modulation of community composition and the functions of cecal microbiota in white-feathered broilers.

## 1. Introduction

In recent years, chicken has emerged as the predominant protein source worldwide due to its increasing demand in the poultry industry, surpassing beef and pork. However, both the intensification of chicken production practices and the ban on antibiotic usage have presented challenges and vulnerabilities for broiler breeders and enterprises [1,2]. Consequently, scholars and experts in livestock production and poultry science have been actively exploring alternative strategies to effectively address these issues, given the escalating demand for antibiotic-free meat products. Feed additives such as antibacterial agents, probiotics, and prebiotics have been extensively administered to improve the growth performance of birds. Among these products, there is a growing interest in characterizing probiotics, which have been shown to improve immunity, antioxidant status, gut function, health and growth parameters in broiler chickens [3,4].

Bacillus-based probiotics are widely abundant in nature and are commonly employed as a substitute for antibiotics in animal and poultry feed. They have the ability to form spores, which confer resistance against high temperatures, pH, bile, and enzymes in the gastrointestinal tract [5,6]. There is mounting evidence suggesting that probiotic supplementation can sustain intestinal health by modifying the gut microbiota; boosting immune regulation, barrier function, and nutrient digestibility; and ultimately enhancing growth performance [7]. Previous studies have demonstrated that the use of Bacillus-based probiotics as a supplement improved growth performance, immunity, gut homeostasis, and microbiota in various animals such as fish, piglets, and chickens [8,9,10]. Meanwhile, other studies evaluated the effects of *B. amyloliquefaciens* LFB112 and *B. subtilis* on the performance of broilers, and observed that feeding broilers with *B. amyloliquefaciens* at a concentration of 5 × 10^5^ cfu/g resulted in improved body weight gain and feed conversion ratio [11].

In this study, 817 white feather broilers were used as a model to investigate the potential positive impact of *B. subtilis* HC6 supplementation on growth performance, immune function, antioxidant capacity, and intestinal health in broilers, which might provide more potential product and evidence for the application of probiotics in broiler breeding.

## 2. Materials and Methods

### 2.1. Bacterial Preparation

*Bacillus subtilis* HC6 (BS) was isolated from our laboratory, and the bacteria were cultured overnight in Lu-ria-Bertani broth at 37 °C, followed by centrifugation at 5000 rpm for 15 min. The resulting pellet was washed twice with sterile phosphate-buffered saline (PBS; pH = 7.4), and then BS powder was prepared at a concentration of 2.0 × 10^8^ cfu/g.

### 2.2. Birds, Experimental Design, and Diets

A total of 180 healthy one-day-old male white feather broilers chicks with an average initial body weight of 36.20 ± 0.01 g were purchased from Muyuan Animal Husbandry Co., Ltd. (Guangzhou, China), and randomly allocated to two groups, with six replicates of 15 birds per replicate. Birds were fed a basal diet, or fed a basal diet supplemented with BS (5.0 × 10^8^ cfu/kg). Pre-experiments before the formal test and previous research results of the members of the research group determined the dosage of probiotics [12]. The diet was formulated according to the nutrient requirements of white-feathered broiler chickens (Table 1), and nutrients met or exceeded nutrient requirements for chickens (NRC, 1994). This study involved housing chicks in a controlled environment with free access to water and feed. The chicks were subjected to twelve-hour light-dark cycles and maintained at different room temperatures throughout the study period. Specifically, the temperature was maintained at 34 °C for the first week, 32 °C for the second week, and 24 °C for the rest of the study period. The humidity was kept at around 62%.

### 2.3. Sample Collection

On day 21 and 50, after 8 h of feeding deprivation, the body weight and the feed residues within each replicate was recorded, then the growth performance parameters, such as the average daily feed intake (ADFI), average daily gain (ADG), and the ratio of feed to gain (FCR), were calculated on a replicate basis and adjusted for mortality during both different phases (1~21 d and 22~50 d) and the whole experiment (1~50 d).

After weighting at d 21 and 50, one broiler chicken that was closest to the average weight of each replicate was selected for blood and tissue sample collection. Briefly, about 3–5 mL of blood was drawn from the brachial vein and allowed to rest before being centrifuged at 3000× *g* for 15 min at 4 °C, and the extracted serum was then preserved at −80 °C to facilitate subsequent determinations. After blood collection, the selected chickens were sacrificed by exsanguination and tissue samples were collected, including the liver, jejunum, ileum, jejunal mucosa, and cecal contents. Liver tissue samples were uniformly collected from the left-side of each chicken, placed in liquid nitrogen for a period, and eventually stored at −80 °C to aid in determining antioxidant capacity. Finally, the jejunum, jejunal mucosa, and the contents of cecum were methodically gathered and preserved using different techniques. Both jejunal mucosa and jejunal tissue samples were obtained from the middle section of the intestinal segment. The jejunal mucosa was collected by scraping the jejunum using a scalpel. Additionally, the scraped jejunal tissue, measuring approximately 3–4 cm in length, was preserved for further analysis. Specifically, the jejunal mucosa and cecal contents were rapidly cryopreserved in liquid nitrogen before being finally stored at −80 °C. Overall, the rigorous sampling and preservation procedures employed in this study laid a solid foundation for reliable and accurate analyses.

### 2.4. Measured Serum Biomarkers

In this study, we utilized assay kits from Nanjing Jiancheng Bioengineering Institute (Nanjing, China) to detect various serum biomarkers. These included triglycerides, total cholesterol, high-density lipoprotein cholesterol, low-density lipoprotein cholesterol, albumin, uric acid, aspartate transaminase, alanine transaminase, alkaline phosphatase, and lactate dehydrogenase. Additionally, the corresponding diamine oxidase (DAO) assay kit and endotoxin kit were used to measure DAO activity and endotoxin level, respectively. The secretory immunoglobulin A (sIgA) and inflammatory factors, including interleukin-2 (IL-2), interleukin-6 (IL-6), interleukin-10 (IL-10), tumor necrosis factor-α (TNF-α), and transforming growth factor-β (TGF-β) concentrations were determined using a chicken-specific ELISA kit (Nanjing Jiancheng, Bioengineering Institute, Nanjing, China). The assay was performed according to the manufacturer’s instructions (Jiancheng Biomedicine, Shanghai, China).

### 2.5. Serum and Liver Antioxidant Markers

The serum antioxidant index was evaluated using a kit from Nanjing Jiancheng Bioengineering Institute (Nanjing, Jiangsu, China), which was comprised of biomarkers such as total superoxide dismutase (T-SOD), total antioxidant capacity (T-AOC), malondialdehyde (MDA), catalase (CAT), and glutathione peroxidase (GSH-Px). After being thawed on ice, liver samples were weighed and homogenized with nine times normal saline at a weight (g):volume (mL) ratio of 1:9. The preparation of homogenate samples from liver tissue follows the experimental methodology provided in the technical support section of the Nanjing Jiancheng Kit website (http://www.njjcbio.com/contents.asp?cid=2&wid=2&id=507, accessed on 7 December 2022). The resulting mixture was then centrifuged, and the supernatant was collected for subsequent measurements. The serum and liver antioxidant indexes were assessed by measuring T-SOD, T-AOC, MDA, CAT, and GSH-Px biomarkers according to the instructions provided by the manufacturer (Nanjing Jiancheng Bioengineering Institute, Nanjing, China).

### 2.6. Intestinal Tissue Morphology

Approximately 1 cm of jejunum and ileum tissue specimens were preserved in a 4% formaldehyde solution for fixation, and they underwent paraffin embedding. Five serial sections were obtained after tissue dehydration and clearing, with the first four sections rehydrated and stained with hematoxylin and eosin. Tissue sections were observed using an optical microscope with a 10× objective lens connected to a LEICA (DFC290 HD system digital camera, Heilbrugger, Switzerland). Villus height and corresponding crypt depth measurements were taken on straight and relatively intact villi, villus height was measured from the crypt-villus junction to the tip of the villus, and crypt depth was measured from the base of the crypt to the crypt-villus junction. The villus/crypt ratio was calculated by dividing the villus height by the crypt depth.

### 2.7. RNA Extraction, cDNA Synthesis and Quantitative PCR

The jejunal mucosa total RNA was extracted using TransZol reagent (TransGen Biotech, Beijing, China), and the concentration, OD 260/280 ratio, and OD 260/230 ratio were determined with high accuracy using a VWRI732-2534 spectrophotometer (Avantor, Radnor, PA, USA) and a mere 1 µL of extracted RNA solution. Subsequently, 1 µg of RNA underwent meticulous reverse transcription and cDNA synthesis utilizing the RT EasyTMII Kit (Foregene, Chengdu, China) to eliminate any genomic DNA contamination. These primers (the primers are presented in Table 2) were synthesized by Qingke Biological Co., Ltd. (Beijing, China), and cDNA gene expression assays were executed via real-time quantitative PCR (QuantStudio 3, Thermo Fisher Scientific, Shanghai, China) utilizing the Real-Time PCR Easy TM-SYBR Green I kit (Foregene, Chengdu, China). The total volume of PCR reaction system was 20 µL, and PCR cycling conditions were 95 °C for 30 s to denature the cDNA template, followed by 40 cycles at 95 °C for 10 s and 60 °C for 30 s. The primer pairs for specific genes, such as immune-related genes and intestinal barrier related genes, were designed using the NCBI primer tool (https://www.ncbi.nlm.nih.gov, accessed on 17 December 2022), and their specificity was verified. All primer pairs were validated, and they exhibited approximately 100% amplification efficiency. Real-time quantitative PCR using the 2^−∆∆Ct^ method was employed to determine the relative gene expression with β-actin as an internal reference.

### 2.8. 16S Sequencing and Cecal Microbiota Analysis

Total genomic DNA from cecal contents was extracted using the Cetyltrimethylammonium Bromide method (CTAB) method (Sample mixing, thermal lysis, chloroform extraction, ethanol washing, and dissolution). Agarose gel electrophoresis was used to assess the DNA extraction quality, and the DNA was quantified using an ultraviolet spectrophotometer. Two primers (V3-forward: 5′-CCTACGGGNGGCWGCAG′ and V4-reverse: 5′-GACTACHVGGGTATCTAATCC-3′) were designed and synthesized for amplifying the V3–V4 region of the 16S rDNA gene. PCR products were purified using AMPure XT beads (Beckman Coulter Genomics, Danvers, MA, USA) and quantified with Qubit (Invitrogen, Waltham, MA, USA). The purified PCR products were evaluated using the Agilent 2100 Bioanalyzer (Agilent, Santa Clara, CA, USA) and Illumina’s library quantification kit (KapaBiosciences, Woburn, MA, USA). The concentration of the qualified library should be above 2 nM. The NovaSeq 6000 sequencer was used for 2 × 250 bp double-ended sequencing with the NovaSeq 6000 SP Reagent Kit (500 cycles). For the double-ended data obtained by sequencing, the samples were first separated based on barcode information, and the joint and barcode sequences were removed. The following steps were then performed for data cleaning and quality control: 1. Primer sequences were removed and the base sequence of RawData was balanced using cutadapt (v 1.9) software; 2. Paired-end reads were combined into longer tags based on the overlap area using FLASH (v 1.2.8) software; 3. Sequencing reads were quality scanned using the window method, with a default scanning window size of 100 bp. If the average quality value within the window was lower than 20, the read part from the beginning of the window to the 3′ end was cut off using fqtrim software (v 0.94); 4. Sequences with a length less than 100 bp were removed; 5. Sequences with a N (uncertain fuzzy bases) content above 5% after truncation were removed; 6. Chimeric sequences were removed using Vsearch (v 2.3.4) software. The Dada 2 tool was invoked with qiime DADA 2 denoise-paired for length filtering and denoising. The ASV (feature) sequence and ASV (feature) abundance table were obtained, and singleton ASVs were removed. Alpha diversity analysis and beta diversity analysis were conducted based on the obtained ASV (feature) sequence and ASV (feature) abundance table. The alpha diversity analysis assesses domestic diversity using six indexes: observed_species, shannon, simpson, chao1, goods_coverage, and pielou_e. Additionally, four types of distances (unweighted_unifrac, weighted_unifrac, jaccard, bray_curtis) were calculated to evaluate the diversity between habitats (samples/groups). The ASV (feature) sequence file, based on the SILVA database (Release 138, available at https://www.arb-SILVA.de/documentation/release138/, accessed on 27 December 2022), was used to annotate the NT-16s database for species. The abundance of each species in each sample was then analyzed using the ASV (feature) abundance table. Spearman’s correlations were analyzed using the R packages heatmap (R 3.6.3). To determine the possible functional profile of the cecal microbiota affected by dietary treatments, we utilized PICRUSt2 (https://github.com/picrust/picrust2, accessed on 7 January 2023), an improved version of the original method. The predicted Kyoto Encyclopedia of Genes and Genomes (KEGG) orthologs were then categorized into Level-2 functional categories.

### 2.9. Statistical Analysis

Independent sample *t*-tests were used for all analyses, and the data were reported as mean ± standard deviation (SD). Statistical analysis and result presentation utilized SPSS software version 22.0 (IBM Corp., Armonk, NY, USA) and GraphPad Prism version 8 (GraphPad Software, Inc., San Diego, CA, USA), respectively. Statistical significance was defined as a *p* value < 0.05 (*p* < 0.05).

## 3. Results

### 3.1. Growth Performance

The growth performance of the broilers is summarized in Table 3. Compared with the control group, the BS group decreased the feed conversion ratio of chickens during periods of 21–50 d (*p* = 0.015) and 1–50 d (*p* = 0.020). Nonetheless, body weight (BW), average daily gain (ADG), and average daily feed intake (ADFI) were comparable between the two groups across all stages (*p* > 0.05).

### 3.2. Intestinal Morphology

Table 4 details a significant rise in the height of jejunal (*p* = 0.001) and ileal (*p* = 0.005) villi, along with an increased villus-to-crypt ratio (*p* = 0.011) in the ileum on day 50 in broilers of the BS-administered group compared to those of the basal diet group.

### 3.3. Jejunal Mucosal Barrier Function

As presented in Figure 1, the effects of BS on the mRNA expression of tight junction proteins, as well as the activity of DAO and the endotoxin of broilers at 21 and 50 days of age, were investigated. The results showed that the mRNA expression levels of ZO-1, Claudin-1, and Occludin were significantly higher in the BS group than those in the CON group on both day 21 and day 50 (*p* < 0.05). However, there was no significant difference in MUC-2 levels between the two groups at either stage (*p* > 0.05). Moreover, serum endotoxin levels and DAO activity were measured to assess intestinal mucosal damage. Broilers in the BS group displayed a significant decrease in endotoxin levels on day 21 and DAO activity on day 50 compared to those in the CON group (*p* < 0.05).

### 3.4. Immune Factor

The levels of serum immune factors for both dietary interventions at days 21 and 50 are listed in Table 5. Compared to the CON group, dietary BS supplementation decreased serum levels of IL-6 (*p* = 0.024) on day 21 and TNF-α (*p* = 0.040) on day 50. However, there was no significant difference in remaining parameters. Table 5 also portrays the impact of BS on immunomodulatory factors in the jejunal mucosa of broilers. Compared with the control group, the addition of BS resulted in an upregulation of IL-10 (*p* = 0.009), IL-22 (*p* = 0.019), and IFN-γ (*p* = 0.035) at the mRNA expression levels on day 21, as well as a downregulation of IL-22 (*p* = 0.001) and IFN-γ (*p* = 0.049) on day 50.

### 3.5. Serum and Liver Antioxidant Index

As shown in Table 6, compared with the CON group, supplementation with BS resulted in significant increases in the serum T-AOC (*p* = 0.016) level, as well as in serum CAT (*p* = 0.006) and T-SOD (*p* = 0.030) activities after 21 days. Moreover, the level of T-AOC (*p* = 0.039), and activities of CAT (*p* = 0.031), T-SOD (*p* = 0.026), and GSH-Px (*p* = 0.003) in serum were elevated after 50 days of treatment with BS. The results also demonstrated that the hepatic activities of T-SOD (*p* = 0.019) and GSH-Px (*p* = 0.006) were significantly increased after 21 days of BS supplementation compared to the control group. Similarly, after 50 days, the level of T-AOC (*p* = 0.029), and hepatic activities of CAT (*p* = 0.034), T-SOD (*p* = 0.025) and GSH-Px (*p* = 0.047) were also significantly increased in the BS supplementation group compared to the control group. In contrast, no significant differences in MDA (*p* > 0.05) content were observed between the different two phases.

### 3.6. Serum Biochemical Index

Table 7 shows the effects of BS on serum biochemical index on 21st and 50th day. The results revealed that, compared to broilers in the CON group, those in the BS group manifested a significant increase in triglyceride content (*p* = 0.004) and aspartate aminotransferase (*p* = 0.043) activity, coupled with a significant decrease in albumin (*p* = 0.049) level on day 21 and a significant increase in HDL-C (*p* = 0.010) level on day 50.

### 3.7. Gut Microbiota

To assess the influence of BS supplementation on the cecal microbiota of broilers, bacterial 16S rDNA amplicon sequencing was performed on the gut microbiota. Table 8 shows that supplementation with BS did not significantly impact the alpha diversity of the bacterial community. As illustrated in Figure 2A,B, the principal coordinate analysis (PCoA) of the weighted UniFrac distances demonstrated distinct segregation between the two groups on day 50, with the first principal component (PCoA1) accounting for 20.96% and the second principal component (PCoA2) accounting for 14.61%, but there was no significant separation on day 21.

The compositions of the top 20 phyla and top 30 genera in the cecal contents of broilers are presented in Figure 2C,D. The dominant phyla were identified as *Firmicutes* on day 21, whereas the dominant phyla were *Firmicutes* and *Bacteroidota* on day 50. There were no significant differences between the CON and BS groups in terms of the abundance of *Firmicutes* and *Bacteroidetes*, or the *Bacteroidetes*/*Firmicutes* ratios in the two stages (Figure 2C). At the genus level, the composition of the microbiota differed among the two groups at different phases (Figure 2D). Compared with the CON group, BS group increased the abundance of *Bacteroidales_unclassified* (*p* = 0.001) and Olsenella (*p* = 0.001), while decreasing the abundance of *Alistipes* (*p* = 0.035). As shown in Figure 2E,F, the effect size measurements (LEfSe) analysis identified 14 and 16 biomarkers with linear discriminant analysis (LDA) scores greater than three in 21 days and 50 days, respectively. Furthermore, two bacterial taxa, namely *Bacteroidaceae* (family) and *Bacteroides* (genus), were found to be enriched in the BS group on day 21, whereas *Clostridiales_unclassified* (family), *Clostridiales_unclassified* (species), *Clostridiales_unclassified* (genus), *Lachnoclostridium* (genus), *Lachnoclostridium_sp_* (species), *Butyricicoccaceae* (family), *Butyricicoccus* (genus), *Butyricicoccus_unclassified* (species), *Anaerotruncus* (genus), *Anaerotruncus_unclassified* (species), *Eubacterium__brachy_group* (genus), and *Eubacterium_brachy_group_unclassified* (species) were enriched in the CON group. Furthermore, on day 50, ten bacterial taxa were found to be enriched in the BS group, including *Corynebacteriaceae* (family), *Corynebacterium* (genus), *Corynebacteriales* (order), *Corynebacterium_unclassified* (species), *Bacteroidetes* (phylum), *Bacteroidales_unclassified* (family), *Bacteroidia* (class), *Bacteroidales_unclassified* (genus), *Bacteroidales* (order), and *Bacteroidales_unclassified* (species), wheseas *Bacteroidales* (order), *Bacteroidia* (class), *Bacteroidota* (phylum), *Rikenellaceae* (family), *Alistipes* (genus), and *Alistipes_unclassified* (species) were abundant in the CON groups.

### 3.8. Spearman Correlation Analysis of Cecal Microbiota

As shown in Figure 3, the top 20 most abundant species at the genus level were selected, and Spearman correlation analysis was performed between bacterial taxa and differential indicators to explore potential links, such as FCR, antioxidant parameters (CAT, T-SOD, T-AOC and GSH-Px), inflammation biomarkers (IFN-γ and IL-22), and intestinal mucosal barrier function (endotoxin, DAO, ZO-1, Occludin and Claudin-1). There were two distinct groups based on the strong correlation between the genera and the parameters. Group 1, including *Bacteroidales_unclassified* and *Olsenella*, was positively correlated with T-SOD, T-AOC, GSH-Px, ZO-1, Occludin, and Claudin-1, and was negatively correlated with FCR, DAO, *IFN-γ*, and *IL-22*. Group 2, including *Clostridium*, *Alistipes*, and *Incertae_Sedis*, was positively correlated with FCR, DAO, IFN-γ, and IL-22, and negatively correlated with T-SOD, T-AOC, GSH-Px, ZO-1, Occludin, and Claudin-1.

### 3.9. Predicted Microbial Functional Analysis by PICRUSt2

PICRUSt 2 microbial functional analysis results determined that BS treatment significantly (*p* < 0.05) enriched six metabolic pathways, including xenobiotics biodegradation and metabolism, carbohydrate metabolism, energy metabolism, signaling molecules and interaction, the digestive system, transport, and catabolism, and downregulated nine metabolic pathways, namely metabolic diseases, the excretory system, lipid metabolism, neurodegenerative diseases, translation, the nervous system, genetic information processing, environmental adaptation, and transcription.

## 4. Discussion

The feed conversion ratio (FCR), an extensively employed metric, reflects the efficiency of feed utilization and production benefits in animal husbandry. The results of this study showed that supplementation with BS increased the feed conversion rate from 21 to 50 days, and throughout the trial period. In line with our results, a previous study pointed out that the introduction of *Bacillus* TL to poultry feed resulted in a significant increase in broiler weight at 21 days and a decrease in feed conversion ratio [13]. Furthermore, Spearman correlation analysis also indicated that the genus *Bacteroidales_unclassified* was negatively correlated with FCR, indicating that *Bacteroidales_unclassified* could improve the efficiency of feed utilization. This may be ascribed to *Bacteroides* being frequently involved in the decomposition of polysaccharides, especially starch and glucan [14]. Taken together, these findings indicate that the administration of BS supplements can decrease FCR, potentially attributable to the influence on cecal microorganisms. Prior research has documented that dietary supplementation with *B. subtilis* probiotics did not affect broiler growth performance during the d 1 to 21 period; instead, it increased average daily gain (ADG) and feed intake, as well as decreased FCR from day 22 to day 42 compared to the control group [15]. In contrast, other studies reported that *Bacillus amyloliquefaciens* and *Saccharomyces cerevisiae* feed supplements did not significantly enhance growth performance at the age of nine weeks, but increased growth performance at the age of 11 weeks in red-feathered native chickens [16]. However, no significant changes in body weight or ADG were observed herein. This discrepancy could be ascribed to multiple factors, including differences in strains, age, breed, testing period, and experimental conditions.

Maintaining a proper balance of intestinal microorganisms is essential for ensuring the health and performance of animals at every stage of their growth [17]. While various segments of the gastrointestinal tract serve distinct functions in host organisms, the cecum houses the most heterogeneous microbial populations, which play a pivotal role in energy extraction and nutrient breakdown, ultimately influencing the intestinal well-being and growth efficiency of chickens [18,19]. Consistent with the findings of previous research, our results demonstrated that *Firmicutes* and *Bacteroidota* were the predominant bacterial phyla [20,21]. Compared with the CON group, the BS group increased the abundance of Bacteroidales_unclassified and *Olsenell*, while decreasing the abundance of *Alistipes*. Moreover, Lefse results showed that *Bacteroides* was the biomarker after BS addition at 21 d, and Bacteroidales_unclassified at 50 d (Figure 2D–F). *Bacteroides* undergo anaerobic respiration to generate acetic acid, isovaleric acid, and succinic acid, which are released into the colon to exert various metabolic effects [22]. *Bacteroidaceae* have the ability to stimulate cytokine production, and an abundance of *Bacteroidaceae* operational taxonomic units (OTUs) may be associated with elevated levels of peripheral cytokines [23]. *Bacteroidales_unclassified* is associated with multiple health-promoting mechanisms, including stimulating the production of SCFAs, regulating the immune system, and enhancing the growth of probiotic microorganisms [24]. These effects might be linked to the expression of ZO-1 and Claudin-1, which contribute to the integrity and stability of the intestinal mucosal barrier. In addition, *Bacteroidale* may stimulate the intestinal immune system and various important metabolic pathways by activating signaling pathways in intestinal epithelial cells [25], including mechanisms such as T-SOD and GSH-Px-related pathways. *Bacteroidales* can also reduce the number and activity of inflammatory cells in the intestinal tract, such as TH1 cells and macrophages, in order to minimize the synthesis of inflammatory mediators [26]. Furthermore, *Alistipes* infections are closely associated with intestinal barrier function and systemic inflammation [27]. This may account for the lower DAO activity in the BS group compared to that in the CON group. In summary, BS increased the abundance of *Bacteroides* and *Olsenell*, which possess the ability to synthesize metabolites such as short-chain fatty acids that can exert antioxidative effects and activate anti-inflammatory signaling pathways.

Moreover, the results showed that BS had no effect on the α-diversity index, but significantly improved the β-diversity index of cecal microbiota. The predictive microbial functional analysis of microbiota using PICRUSt2 indicated that there was an increase in xenobiotics biodegradation and metabolism, carbohydrate metabolism, energy metabolism, signaling molecules and interaction, the digestive system, transport, and catabolism in the BS group, implying improvements in the digestibility of carbohydrates and proteins, as well as a potential amelioration in the homeostatic mechanism (Figure 4). Of note, carbohydrate-metabolism-related pathways encompass a range of metabolic processes related to energy metabolism, such as those involving glycogen, glucose, disaccharides, pentose phosphate, and glycosaminoglycan [28]. The addition of live BS to the diet resulted in changes to the composition and diversity of the intestinal microbiota. Specifically, there was an increase in pathways related to carbohydrate metabolism and energy metabolism [29]. Altogether, these findings are consistent with the results of the current study.

In the small intestine, over 90% of nutrient digestion and absorption occur, underscoring the paramount importance of its epithelial integrity in sustaining gut health [30]. Villus height and crypt depth are known to play a vital role in nutrient absorption and digestion. Better feed digestion and nutrient absorption capability may also be indirectly related to the immunostimulatory and antioxidative effects of *Bacillus* strains, which can significantly improve the intestinal morphology of the intestinal tract. Moreover, the immune-modulatory action enhances the mucosal barrier integrity, and the intestinal mucosal structure increases the villus height and the villus height to crypt depth ratio [31,32]. Prior research has demonstrated that probiotics led to increases in villus height and the villus height to crypt depth ratio, and decreases in the crypt depth of broiler chicken [33,34]. Similarly, BS increased the villus height of the jejunum and ileum, as well as the ileal villus height to crypt depth ratio on day 50, but had no significant effect on intestinal morphology on day 21 (Table 4). We speculate that BS supplementation exerts a positive impact on nutrient absorption and digestion by increasing intestinal morphology, which is partly reflected by the lower FCR during periods of 21–50 d and 1–50 d.

The intestinal epithelial barrier plays a regulatory role in mucosal homeostasis and defense against foreign invasion [35]. Intestinal epithelial cells rely on tight junctions (TJs) and adhesion junctions (AJs) to execute their barrier function [36]. Impaired intestinal physical barrier function is characterized by increased intestinal permeability and the downregulated expression of AJs and TJs. Furthermore, the endotoxin level and DAO activity are positively correlated with intestinal permeability [37]. Noteworthily, elevated intestinal permeability can compromise the integrity of the intestinal barrier, facilitating the translocation of bacteria and other detrimental agents into the bloodstream, thereby inciting inflammation and other health complications. Our results showed that BS significantly reduced the endotoxin level on day 21 and decreased DAO activity on day 50, accompanied by upregulated mRNA expression levels of ZO-1, Occludin, and Claudin-1 in the jejunal mucosa (Figure 1). In agreement with our findings, some studies have found that supplementation with probiotics reduced the serum endotoxin level and DAO activity in mice and *E. coli*-infected birds [38,39]. Dietary supplementation with BL-S6 significantly elevated both the mRNA and protein levels of tight junction proteins (ZO-1 and Occludin) in the intestinal epithelial cells of weaned piglets [20]. Additionally, *B. licheniformis* was found to upregulate intestinal MUC-2 expression in chickens and pigs, emphasizing its role in fortifying the mucus layer as a chemical barrier against bacterial infections [40,41,42]. However, our results indicate that *Bacillus subtilis* does not affect MUC-2 expression, which is consistent with previous findings [43]. Moreover, Spearman correlation analysis also indicated that the genus *Bacteroidales_unclassified* was positively correlated with the mRNA expression levels of tight junction proteins ZO-1, Occludin, and Claudin-1, which are essential for preserving intestinal integrity and barrier function. Based on the results of this study, we can gain a better understanding of the beneficial effects of BS on the intestinal epithelial barrier function of chickens.

Immunosuppression in the poultry industry, attributed to factors such as over-crowding, stress, and pathogenic infections, can significantly compromise growth performance, decrease the quality of poultry products, and result in diminished economic returns [44,45,46]. Pro-inflammatory cytokines (such as IFN-γ and TNF-α) and anti-inflammatory cytokines (such as IL-10) assume central roles in bolstering cellular immunity and preserving immune homeostasis during the process of inflammation [47]. Hence, it is crucial to establish and sustain adequate immunological function to promote healthy growth in broilers [48]. Numerous studies have reported the ability of probiotics to modulate host immune function, and they have thus been extensively utilized in both animal and human populations to boost disease resistance capacity [49,50,51]. Our results revealed that BS supplementation increased IL-6 levels at 21 d, and concurrently decreased TNF-α at 50 d in serum, as well as upregulated the mRNA expression of IL-10, IL-22, and IFN-γ in the jejunal mucosa, collectively signifying that probiotic *B. subtilis* might serve as a promising immune modulator for mediating the intestinal immune function of broilers. Studies conducted on mice have shown that certain Bacillus strains possess immunomodulatory properties. In particular, *B. subtilis* strains (*B. subtilis* B10, *B. subtilis* BS02, and *B. subtilis* (natto) B4 spores) have been found to activate macrophages and induce the production of pro-inflammatory cytokines [52]. However, its efficacy may vary owing to the dynamic equilibrium of the immune system and other factors such as environment and nutrition. Furthermore, Spearman correlation analysis also indicated that the levels of IL-22 and IFN-γ were negatively correlated with the *Bacteroidales_unclassified* and *Olsenella* genera, but positively correlated with the *Alistipes* genus. This observation offers insights into the underlying rationale for the decrease in the proportion of immune-regulatory signaling molecules in the jejunal mucosa of 50-day-old broilers. Nevertheless, further research is warranted to comprehensively elucidate this phenomenon. Hence, these results suggest that BS supplementation modulates immune function in broilers by improving gut microbiota composition.

Apart from their immunomodulatory effect, certain Bacillus strains have been found to exhibit antioxidant activities, which are hypothesized to be facilitated through the synthesis of antioxidant enzymes that confer protection to the host against oxidative stress [53]. Among the antioxidant enzymes, Bacillus spp. are known to synthesize superoxide dismutase (SOD), glutathione peroxidase (GSH-Px), and glutathione reductase (GR). As is well documented, free radicals can damage the lipids in cell membranes and generate MDA, which affects polyunsaturated fatty acids found in biological cell membranes, further boosting lipid peroxidation, which in turn exacerbates oxidative stress [54]. Antioxidant levels reflect the body’s ability to scavenge oxygen-free radicals and maintain homeostasis. T-AOC is typically to estimate antioxidative potential, while SOD and GSH-Px are regarded as crucial enzymatic antioxidants. In addition, GSH acts as an essential nonenzymatic antioxidant that eliminates free radicals [55]. The present study revealed that the inclusion of *Bacillus subtilis* resulted in elevated levels of antioxidant biomarkers, namely T-AOC, T-SOD, CAT, and GSH-Px, in both serum and hepatic samples at two distinct stages. Earlier research outcomes parallel our results in that B. subtilis fmbJ can increase the levels of glutathione (GSH), GR, glutathione peroxidase (GSH-Px), and SOD activity in broiler serum and liver, leading to decreased mitochondrial ROS content in the liver [15]. Likewise, the findings of multiple studies are consistent with those of our trial [56,57]. Furthermore, Spearman correlation analysis also indicated that the abundance of the *Bacteroidales_unclassified* genus was positively correlated with T-SOD and GSH-Px activities, potentially accounting for why the higher antioxidant activity in the BS group was higher than that in the CON group. In summary, these outcomes signify that Bacillus may play a beneficial role in oxidative defense, owing to its inherent antioxidant activity.

## 5. Conclusions

The results of the present study suggested that dietary supplementation with 5 × 10^8^ cfu/kg *Bacillus subtilis* HC6 improves the feed conversion ratio and antioxidant capacity in serum and liver, increases the levels of high-density lipoprotein and aspartate aminotransferase, and decreases the activities of diamine oxidase and endotoxin. Likewise, dietary *Bacillus subtilis* upregulated the mRNA expression of tight junction proteins in the jejunal mucosa, and changed the cecal microflora composition of broilers. Meanwhile, Spearman correlation analysis identified a correlation between cecal microbiota composition and FCR, serum differential metabolites or differential gene expression in the jejunal mucosa, which might be attributed to the increased *Bacteroidales_unclassified* and *Olsenella* abundance, and decreased *Alistipes* abundance in broilers. The aforementioned results offer novel insights into the utilization of *Bacillus subtilis*, specifically by placing emphasis on its influence on the composition, functional capacity, and diversity of gut microbiota. Further research endeavors are necessary to clarify the potential mechanisms underlying host-microbe interactions in metabolism and physiology, ultimately holding the potential to improve the overall performance and well-being of broilers.

## Figures and Tables

**Figure 1 animals-13-02915-f001:**
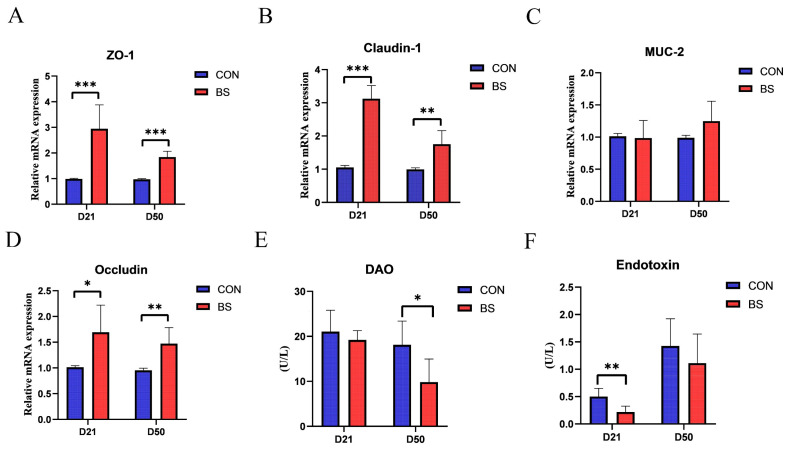
Effects of BS on tight junction protein mRNA expression, serum DAO activity, and endotoxin activity of broilers on day 21 and day 50. (**A**) ZO-1; (**B**) MUC-2; (**C**) Claudin-1; (**D**) Occludin; (**E**) DAO; (**F**) Endotoxin; * *p* < 0.05, ** *p* < 0.01, and *** *p* < 0.001 compared with CON group. CON, control; BS, *Bacillus subtilis* HC6; DAO, diamine oxidase.

**Figure 2 animals-13-02915-f002:**
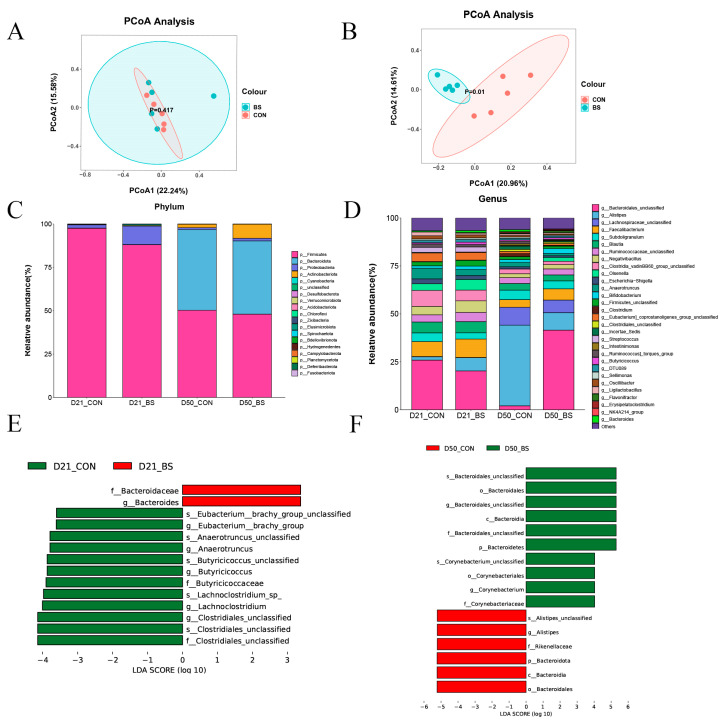
Relative abundance of the chicken cecal microbiota; (**A**) D 21 principal coordinate analysis plot; (**B**) D 50 principal coordinate analysis plot; (**C**) Phylum-level taxonomic composition of the cecal microbiota; (**D**) Genus-level taxonomic composition and relative abundance of the cecal microbiota; (**E**) Lefse identified the most differential genera on 21 days; (**F**) Lefse identified the most differential genera on 50 days. CON, control; BS, *Bacillus subtilis* HC6.

**Figure 3 animals-13-02915-f003:**
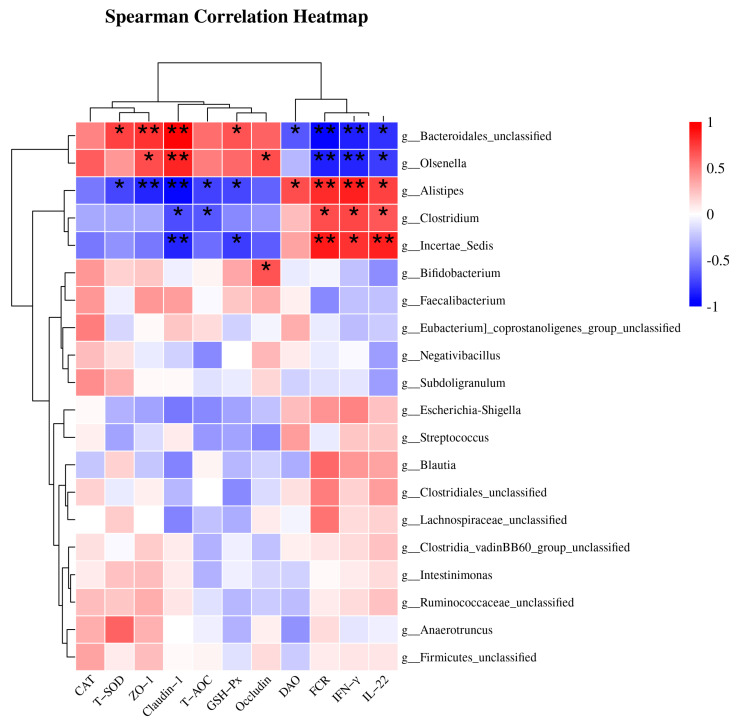
This heatmap displays the results of Spearman correlation analysis between specific bacterial taxa, FCR, and biochemical parameters. Statistically significant associations (*p* < 0.05) were observed between all indicators and bacterial genera. The red blocks indicate positive correlations, while the blue blocks indicate negative correlations. The intensity of the colors reflects the strength of the correlations. * *p* < 0.05 and ** *p* < 0.01 indicate significant differences compared to the CON group; DAO, diamine oxidase; CON, control; BS, *Bacillus subtilis* HC6.

**Figure 4 animals-13-02915-f004:**
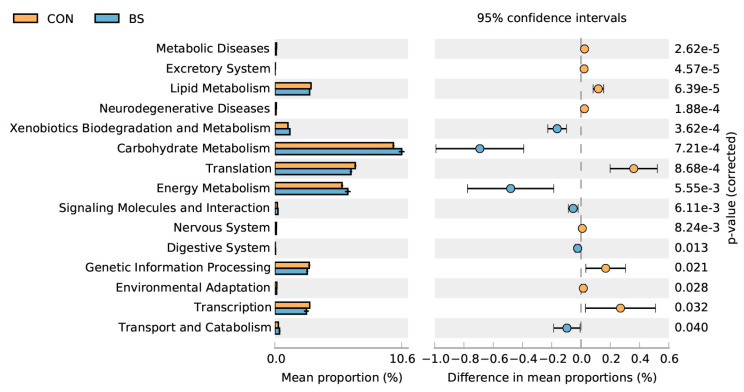
Comparison of predicted pathway abundances on day 50 between two groups by statistical analysis of taxonomic and functional profiles (STAMP). CON, control; BS, *Bacillus subtilis* HC6.

**Table 1 animals-13-02915-t001:** Composition and nutrient content of experimental diets.

Diets	Starter (1–20 Days)	Finisher (21–50 Days)
Ingredients (g/kg)		
Corn	58.75	57.66
Soybean meal (43% CP)	27.70	27.21
Corn gluten meal (60% CP)	5.00	5.00
Limestone	1.43	1.22
Calcium hydrogen phosphate (16.5%)	1.10	1.04
L-lysine Sulfate (70%)	0.66	0.39
DL-methionine (98.5%)	0.33	0.21
NaCl	0.28	0.32
L-Threonine	0.16	0.06
Peanut meal	3.00	0
Choline chloride (50%)	0.08	0.08
Premix ^1^	0.35	0.35
Phytase (20,000 IU)	0.01	0.01
Sodium humate	0	0.15
Lard	1.15	6.30
Total (%)	100	100
Nutrient content ^2^		
ME (Kcal/kg)	2941.61	3231.28
CP (g/kg)	21.92	19.61
CEE (g/kg)	3.65	8.87
CF (g/kg)	2.51	2.33
Ca (g/kg)	0.90	0.80
Total P (%)	0.57	0.54
Lys (%)	1.48	1.21
Met (%)	0.64	0.52
Cys (%)	0.30	0.27
Met + Cys (%)	0.94	0.78
L-Threonine	0.92	0.79

^1^ Premix is provided for each kilogram of diet: vitamin A, 12,000 IU; vitamin D3, 3000 IU; vitamin E, 10 IU; vitamin K3, 2 mg; vitamin B1, 1 mg; vitamin B2, 3 mg; vitamin B6, 2 mg; vitamin B12, 0.01 mg; niacin, 20 mg; calcium pantothenate, 4 mg; biotin, 0.05 mg; folic acid, 0.5 mg; Mn, 100 mg; Fe, 100 mg; Zn, 80 mg; Cu, 20 mg; I, 3 mg; and Se, 0.5 mg. ^2^ Nutrient levels were calculated values. L-, L-isomer; DL-, DL-isomer.

**Table 2 animals-13-02915-t002:** The primer sequence of the gene.

Gene	GenBank	Sequence (5′–3′)	TM °C
*β-actin*	NM_205518.2	F: CATTGTCCACCGCAAATGCT R: AAGCCATGCCAATCTCGTCT	57.2
*IL-6*	NM_204628.2	F: CAGGACGAGATGTGCAAGAA R: TAGCACAGAGACTCGACGTT	56.6
*IL-10*	NM_001004414.4	F: CGCTGTCACCGCTTCTTCA R: CTTTGTTCTCATCCATCTTCTC	57.0
*IL-1β*	XM_046931582.1	F: CGACATCAACCAGAAGTGCTT R: GTCCAGGCGGTAGAAGATGA	56.8
*IL-22*	NM_001199614.1	F: GCCCTACATCAGGAATCGCA R: TCTGAGAGCCTGGCCATTTC	57.8
*IL-17A*	NM_204460.2	F: GAAGGTGATACGGCCAGGAC R: TGGGTTAGGCATCCAGCATC	56.8
*IFN-β*	NM_001024836.2	F: TGCAACCATCTTCGTCACCA R: GGAGGTGGAGCCGTATTCT	56.68
*IFN-γ*	NM_205149.2	F: ACACTGACAAGTCAAAGCCGC R: AGTCGTTCATCGGGAGCTTG	58.6
*TNF-α*	XM_046927265.1	F: TGTGTATGTGCAGCAACCCGTAGT R: GGCATTGCAATTTGGACAGAAGT	57.8
*MUC-2*	XM_040673077.2	F: TTCATGATGCCTGCTCTTGTG R: CCTGAGCCTTGGTACATTCTTGT	58.0
*ZO-1*	XM_046925214.1	F: CTTCAGGTGTTTCTCTTCCTCCTC R: CTGTGGTTTCATGGCTGGATC	56.6
*Claudin-1*	NM_001013611.2	F: GGTATGGCAACAGAGTGGCT R: CAGCCAATGAAGAGGGCTGA	57.0
*Occludin*	XM_046904540.1	F: GATGGACAGCATCAACGACC R: CATGCGCTTGATGTGGAAGA	58.0

**Table 3 animals-13-02915-t003:** Effects of BS on growth performance in broilers.

Items	CON	BS	*p*-Value
1 d BW, g	36.20 ± 0.01	36.20 ± 0.01	-
21 d BW, g	535.28 ± 21.96	543.06 ± 14.35	0.635
50 d BW, g	2040.00 ± 115.33	2092.67 ± 55.15	0.515
1–21 d			
ADG, g/d	24.95 ± 1.10	25.34 ± 0.72	0.634
ADFI, g/d	35.20 ± 1.88	34.79 ± 0.86	0.746
FCR	1.41 ± 0.02	1.37 ± 0.04	0.180
21–50 d			
ADG, g/d	51.88 ± 3.32	53.44 ± 1.42	0.499
ADFI, g/d	109.63 ± 6.66	103.03 ± 2.30	0.093
FCR	2.12 ± 0.08	1.93 ± 0.02	0.015
1–50 d			
ADG, g/d	40.83 ± 2.46	41.90 ± 1.23	0.537
ADFI, g/d	79.25 ± 3.52	75.18 ± 1.69	0.145
FCR	1.94 ± 0.06	1.80 ± 0.03	0.020

BW, body weight; ADG, average daily gain; ADFI, average daily feed intake; FCR, feed conversion ratio; CON, control; BS, *Bacillus subtilis* HC6.

**Table 4 animals-13-02915-t004:** Effects of treatments on jejunal and ileal morphology.

Items	CON	BS	*p*-Value
jejunum			
21 d Villus height, µm	1029.94 ± 45.42	1062.01 ± 120.92	0.594
21 d Crypt depth, µm	111.80 ± 12.99	120.75 ± 15.37	0.349
21 d V/C	9.28 ± 0.85	9.03 ± 2.42	0.833
50 d Villus height, µm	1090.97 ± 17.91	1204.35 ± 38.96	0.001
50 d Crypt depth, µm	111.59 ± 13.17	118.24 ± 12.82	0.442
50 d V/C	9.87 ± 0.97	10.29 ± 1.04	0.557
ileum			
21 d Villus height, µm	589.94 ± 42.97	623.35 ± 45.29	0.266
21 d Crypt depth, µm	81.80 ± 5.51	79.70 ± 2.57	0.463
21 d V/C	7.22 ± 0.47	7.82 ± 0.46	0.079
50 d Villus height, µm	780.74 ± 53.21	873.91 ± 14.20	0.005
50 d Crypt depth, µm	115.59 ± 4.82	112.89 ± 5.56	0.435
50 d V/C	6.77 ± 0.60	7.75 ± 0.29	0.011

V/C, Villus height/Crypt depth; CON, control; BS, *Bacillus subtilis* HC6.

**Table 5 animals-13-02915-t005:** Effects of BS on the serum and immune jejunal mucosa factors of broilers.

Items	CON	BS	*p*-Value
serum			
D 21			
IL-2, ng/L	36.85 ± 11.64	44.90 ± 13.98	0.351
IL-6, ng/L	61.11 ± 25.48	31.17 ± 13.88	0.024
TNF-α, ng/L	132.61 ± 14.63	139.84 ± 15.07	0.463
TGF-β, ng/L	3110.92 ± 247.55	2899.92 ± 185.31	0.166
IL-10, ng/L	11.40 ± 5.91	12.34 ± 3.99	0.775
sIgA, ng/L	234.49 ± 38.47	255.39 ± 51.42	0.487
D 50			
IL-2, ng/L	21.71 ± 5.24	25.50 ± 7.64	0.388
IL-6, ng/L	90.20 ± 14.48	87.11 ± 13.12	0.733
TNF-α, ng/L	191.10 ± 14.90	167.30 ± 15.75	0.040
TGF-β, ng/L	2630.09 ± 264.19	2448.74 ± 152.73	0.221
IL-10, ng/L	4.54 ± 1.72	4.27 ± 2.99	0.869
sIgA, ng/L	169.85 ± 12.65	173.92 ± 28.14	0.776
jejunal mucosa			
D 21			
IL-6	1.09 ± 0.10	1.23 ± 0.54	0.636
IL-10	1.04 ± 0.13	1.88 ± 0.42	0.009
IL-1β	1.04 ± 0.03	1.73 ± 0.68	0.089
IL-22	1.10 ± 0.09	1.53 ± 0.26	0.019
IL-17A	1.02 ± 0.04	1.34 ± 0.72	0.418
IFN-γ	1.05 ± 0.09	1.78 ± 0.53	0.035
TNF-α	1.05 ± 0.07	1.40 ± 0.64	0.312
D 50			
IL-6	1.02 ± 0.05	0.63 ± 0.35	0.069
IL-10	1.02 ± 0.05	1.01 ± 0.20	0.932
IL-1β	1.03 ± 0.04	0.64 ± 0.36	0.092
IL-22	1.07 ± 0.10	0.33 ± 0.20	0.001
IL-17A	1.03 ± 0.04	0.60 ± 0.38	0.067
IFN-γ	1.04 ± 0.09	0.64 ± 0.32	0.049
TNF-α	1.06 ± 0.07	1.35 ± 0.36	0.166

CON, control; BS, *Bacillus subtilis* HC6.

**Table 6 animals-13-02915-t006:** Effects of BS on serum and liver antioxidant index of broilers.

Items	CON	BS	*p*-Value
serum			
D 21			
MDA, nmol/mL	5.01 ± 1.50	3.93 ± 1.02	0.787
T-AOC, U/mL	3.84 ± 1.45	6.38 ± 1.16	0.016
CAT, U/mL	14.60 ± 2.61	21.41 ± 2.92	0.006
T-SOD, U/mL	28.52 ± 9.76	42.87 ± 7.28	0.030
GSH-Px, umol/L	708.93 ± 235.55	946.41 ± 357.81	0.250
D 50			
MDA, nmol/mL	3.70 ± 1.11	2.84 ± 0.53	0.156
T-AOC, U/mL	3.70 ± 1.13	7.39 ± 3.14	0.039
CAT, U/mL	3.80 ± 0.77	5.16 ± 0.87	0.031
T-SOD, U/mL	43.95 ± 7.04	53.99 ± 4.33	0.026
GSH-Px, umol/L	1178.90 ± 273.97	1787.62 ± 230.15	0.003
liver			
D 21			
MDA, nmol/mgprot	4.73 ± 1.13	4.35 ± 1.39	0.648
T-AOC, U/mgprot	6.49 ± 0.26	7.38 ± 1.04	0.105
CAT, U/mgprot	18.50 ± 6.43	26.49 ± 5.49	0.068
T-SOD, U/mgprot	151.88 ± 34.60	222.66 ± 41.59	0.019
GSH-Px, umol/mgprot	52.14 ± 7.83	73.29 ± 10.26	0.006
D 50			
MDA, nmol/mgprot	5.33 ± 0.69	5.22 ± 0.44	0.772
T-AOC, U/mgprot	5.34 ± 0.31	6.63 ± 1.03	0.029
CAT, U/mgprot	16.99 ± 3.65	27.49 ± 8.41	0.034
T-SOD, U/mgprot	268.32 ± 33.59	350.72 ± 57.95	0.025
GSH-Px, umol/mgprot	63.81 ± 8.16	87.35 ± 20.96	0.047

T-SOD, total superoxide dismutase; T-AOC, total antioxidant capacity; MDA, malondialdehyde; CAT, catalase; GSH-Px, glutathione peroxidase; CON, control; BS, *Bacillus subtilis* HC6.

**Table 7 animals-13-02915-t007:** Effects of BS on serum physiological and biochemical indexes of broilers.

Items	CON	BS	*p*-Value
D 21			
Triglyceride, mmol/L	0.50 ± 0.08	0.66 ± 0.06	0.004
Total cholesterol, mmol/L	4.21 ± 0.47	3.78 ± 0.42	0.121
High-density lipoprotein, mmol/L	9.46 ± 1.15	8.89 ± 1.19	0.423
Low density lipoprotein, mmol/L	1.01 ± 0.13	1.07 ± 0.13	0.477
Albumin, g/L	20.87 ± 1.55	18.03 ± 1.77	0.049
Uric acid, umol/L	188.43 ± 56.90	217.13 ± 42.72	0.346
Lactate dehydrogenase, U/L	571.51 ± 52.76	525.00 ± 69.27	0.220
Alkaline phosphatase, mg/L	609.34 ± 129.37	526.01 ± 82.99	0.214
Aspartate aminotransferase, U/L	27.66 ± 7.30	36.97 ± 6.59	0.043
Alanine aminotransferase, U/L	34.34 ± 7.66	45.43 ± 22.07	0.271
D 50			
Triglyceride, mmol/L	0.75 ± 0.19	0.86 ± 0.20	0.317
Total cholesterol, mmol/L	4.26 ± 0.67	4.75 ± 1.48	0.476
High-density lipoprotein, mmol/L	3.62 ± 1.51	5.65 ± 0.43	0.010
Low density lipoprotein, mmol/L	3.43 ± 0.86	2.97 ± 1.07	0.432
Albumin, g/L	20.87 ± 4.03	20.28 ± 2.84	0.775
Uric acid, umol/L	353.52 ± 72.75	320.46 ± 60.03	0.411
Lactate dehydrogenase, U/L	607.26 ± 25.79	566.99 ± 48.10	0.101
Alkaline phosphatase, mg/L	155.56 ± 29.88	174.18 ± 40.97	0.389
Aspartate aminotransferase, U/L	15.36 ± 3.88	22.65 ± 8.07	0.074
Alanine aminotransferase, U/L	21.12 ± 12.45	13.93 ± 9.18	0.281

CON, control; BS, *Bacillus subtilis* HC6.

**Table 8 animals-13-02915-t008:** Alpha diversity indexes of cecal microbiota.

Items	CON	BS	*p*-Value
D 21			
Chao 1	502.72 ± 124.09	495.09 ± 165.60	0.936
Shannon	6.96 ± 0.30	6.41 ± 1.40	0.411
Simpson	0.98 ± 0.01	0.96 ± 0.04	0.409
Goods_coverage	1.00	1.00	-
Pielou_e	0.81 ± 0.07	0.72 ± 0.12	0.199
Observed_otus	501.80 ± 123.43	494.40 ± 165.47	0.938
D 50			
Chao 1	408.64 ± 51.88	441.90 ± 69.34	0.415
Shannon	4.83 ± 0.42	5.26 ± 0.54	0.206
Simpson	0.84 ± 0.06	0.89 ± 0.07	0.254
Goods_coverage	1.00	1.00	-
Pielou_e	0.56 ± 0.04	0.60 ± 0.08	0.313
Observed_otus	406.60 ± 51.85	440.00 ± 69.27	0.413

CON, control; BS, *Bacillus subtilis* HC6.

## Data Availability

The data presented in this study are available on request from the corresponding author.

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
