# Peer review of "Effects of Dietary Bacillus subtilis HC6 on Growth Performance, Antioxidant Capacity, Immunity, and Intestinal Health in Broilers"

_animals, 2023, doi:10.3390/ani13182915_

Round 1

Reviewer 1 Report

Thanks for the opportunity to review. The paper has some merits, although the novelty is limited and not clearly described. I have appended comments for the authors to address. I would like them to respond to each of them and submit a response note together with the revised manuscript.

A professional English language editor can improve the readability and quality of the manuscript.

Reviewer 2 Report

In this study, researchers determined effects of probiotic supplementation on indicators of health and growth in the small intestine, liver, and serum of chickens. Results suggest a beneficial effect of dietary supplementation in terms of feed efficiency, gut health, and antioxidant capacity in the blood and liver. Despite the promising results, the methods and results were difficult to understand, partly because of a lack of detail and inconsistencies among sections. Specific comments are below:

More information needed on strain of BS used in broiler studies. Where was the strain originally obtained – is there a previous reference for this that can be cited? An internet search did not reveal information on Bacillis subtilis HC6.

Line 74 – At what age were birds received and assigned to treatment?

There is no mention of how chickens were assigned to treatment. Were body weights recorded and were they taken into account when assigning birds to treatment?

The citation for the nutrient recommendations for chickens is unclear. There is no NRC publication for poultry dated 2012.

Line 78 – Would like to see the actual amount of probiotic that was added per kg of feed

Line 81 – not clear what is meant by a “standardized feeding method”. Were they ad lib or restricted? Were there differences in feed intake among treatment groups?

Housing requires more detail – were they caged or floor-penned? What was the “replicate” – a cage, pen or chicken?

Line 92 – body weights are mentioned here, but collection of body weight data is not mentioned elsewhere in methods.

Line 93 – if you selected the bird with the body weight closest to the average BW of the replicate group, how is that a “random” selection for sampling?

Line 94 – wording is unclear. What is meant specifically by “consistency and reliability” with respect to fasting the chickens for 8 hours before sampling?

Lines 93-106 – it is assumed that careful attention is paid to each procedure and that researchers are meticulous. This type of wording implies that other parts of the research were not as meticulous. This type of wording should be removed.

Lines 93-106- How were birds euthanized? Were they euthanized before or after blood collection?

Lines 101-104 – more detail is needed on intestinal collection. Where were samples collected – middle of the segment? How was mucosa collected? Was the “whole jejunum” collected with mucosa or was it the tissue remaining after the mucosa was removed? How much tissue was collected?

Lines 115-118 – more detail is needed on the ELISA kits. Were they chicken specific?

Line 124 – not sure what is meant by “nine times nor-mal saline”

Line 131 – this does not make sense. What is meant by a repetition of 6 birds? Does this mean that 6 birds is one experimental unit?

Line 131- assumed that “artificial neck” is referring to cervical dislocation? Unclear wording. Cervical dislocation is mentioned here but not elsewhere – it should be mentioned earlier in the methods where sample collection is described.

Line 132: Earlier in methods only jejunum is mentioned with respect to histology. If ileum was also collected, this needs to be mentioned earlier.

Lines 133-138 – more detail needed on methods. What type of software was used to measure villus height and crypt depth? How many villi and crypts were measured per section and bird?

Lines 140-141 – repetition with earlier section. Sampling dates mentioned here but not in previous section. If all samples were collected at same time points, it suffices to mention this once where sampling is described earlier in the methods.

Line 140 – wording is unclear with “RNA” as the first word in the sentence.

How was RNA tested for integrity?

Line 147 – again, the word “meticulous” should be deleted

Lines 154-156 – Was amplification efficiency validated for each primer pair? If so, efficiencies should be reported – the delta-delta CT method of quantification assumes 100% efficiency of reference and target genes. Product size seems large for real time PCR – e.g., IL-6 >200bp, IL-1B 298bp

Lines 162-168 – what instrumentation was used for sequencing?

Table 2 – Accession IDs should be reported for each gene

Results – Growth performance measurements are not mentioned in the methods

Why are time points treated individually in results – why not analyze as ANOVA?

Methods and Table 3 – there is no mention of how FCR is calculated. Superscripts in Table 3 not needed since only 2 groups are being compared and it is obvious which is larger than the other. Units are needed in table.

Table 4 – superscripts not needed when only 2 groups are compared.

Figure 1 – caption requires more detail. Need to specify mRNA for PCR measurements and enzyme activity for DAO, otherwise it is implied that all measurements are of the protein.

Line 297 and elsewhere (e.g., Line 421 and rest of discussion): because you did not measure protein levels of these tight junction factors, they should be referred to as “mRNAs” when you mention them. Otherwise, when you describe the “biochemical parameters” it is implied that you are referring to the proteins. In the abstract and discussion it is implied that tight junction proteins were measured - this is not the case and must be corrected.

There is a lot of redundancy in the discussion - avoid repetitive statements and repeating the same results multiple times. 

This paper requires editing of grammar and mechanics.

Round 2

Reviewer 1 Report

Thanks for the revisions, the paper can be acceptable upon moderate language edits.

There is still a need for language edits. A professional native editor can assist to improve the readability and quality of the paper.

Reviewer 2 Report

The authors have addressed all concerns and manuscript is improved.